# Pirarubicin Combination Low-Dose Chemotherapy for Early Infantile Stage MS Neuroblastoma: Case Report

**DOI:** 10.3390/children10050871

**Published:** 2023-05-12

**Authors:** Yutaka Kato, Hiroshi Kawaguchi, Naoki Sakata, Satoshi Ueda, Munehiro Okano, Yuuki Nishino, Masako Ryujin, Yutaka Takemura, Tsukasa Takemura, Keisuke Sugimoto, Satoshi Okada

**Affiliations:** 1Department of Pediatrics, Hiroshima University Graduate School of Biomedical and Health Sciences, Hiroshima-Shi 734-8551, Hiroshima, Japan; hrsmytk@hiroshima-u.ac.jp (Y.K.);; 2Department of Pediatrics, Faculty of Medicine, Kindai University, 377-2 Ohno-higashi, Osaka-Sayama 589-8511, Osaka, Japan; 3Ueda Child Clinic, 591-6 Tsubakihara, Hashimoto 648-0052, Wakayama, Japan; 4Department of Pediatrics, Kaizuka City Hospital, 3-10-20 Hori, Kaizuka 597-0015, Osaka, Japan; 5Department of Pediatrics, Sakai Sakibana Hospital, 2-7-1 Harayamadai, Minami-ku, Sakai 590-0132, Osaka, Japan; 6Department of Pediatrics, Kushimoto Municipality Hospital, 691-7, Sangodai, Kushimoto, Higashimuro 649-3510, Wakayama, Japan

**Keywords:** pirarubicin, low-dose chemotherapy, neuroblastoma, early infant, stage MS

## Abstract

Neuroblastoma (NB) is a neural crest-derived malignant tumor which is diagnosed during infancy in approximately 40% of cases; spontaneous regressions are observed, but there are varying degrees of severity. Treatment is indicated if an infant’s condition is at risk of deterioration. Herein, we report the case of a 42-day-old boy who presented with hepatomegaly and was diagnosed with stage MS NB. A pathological diagnosis of “poorly differentiated neuroblastoma with low mitosis-karyorrhexis index, favorable histology” was made; his tumor cells were hyperdiploid and MYCN was not amplified. Because he had respiratory distress caused by the rapidly evolving hepatomegaly, two cycles of chemotherapy containing vincristine and cyclophosphamide were administered in the second and fourth weeks of admission; however, his abdominal tumor did not shrink. In the sixth week of admission, chemotherapy was revised to pirarubicin and cyclophosphamide, and the tumor began to shrink. After discharge, there was no re-elevation of tumor markers; after 1 year, the hepatomegaly and liver metastases disappeared. During the 5-year follow-up, his growth and development were normal and he progressed without sequelae. A regimen that includes pirarubicin could merit further study in the treatment of early infants with stage MS low-risk NB who are at risk of complications.

## 1. Introduction

Neuroblastoma (NB) is a neural crest-derived malignant tumor that is diagnosed in infancy in approximately 40% of cases [1]. Spontaneous regressions are observed, particularly in infants, and it is generally associated with good outcomes in infants. There are varying degrees of severity, and treatment is indicated if an infant’s condition is at risk of deterioration [2,3].

Stage MS NB classified using the International Neuroblastoma Risk Group (INRG) Staging System is a metastatic disease, usually diagnosed in patients younger than 18 months [4]. The same disease is classified as stage 4S NB (younger than 12 months) using the International Neuroblastoma Staging System (INSS) [5]. The metastases of these children are confined to the skin, liver, and/or bone marrow (BM). [^123^I]-metaiodobenzylguanidine (^123^I-MIBG) scintigraphy should be negative in the bone and BM, and BM tumor involvement should be limited to <10% of the total nucleated cells on smears or in biopsies [4]. Stage MS NB usually has a good prognosis. Patients diagnosed with stage MS NB are usually followed up regularly, or treated using chemo- and surgical therapy, and a therapeutic approach must be determined for each patient [2,3]. In a prospective study, 49 infants with stage 4S NB have received chemotherapy with carboplatin (CBDCA), etoposide (ETP), cyclophosphamide (CY), and doxorubicin (DXR) [2,6]. In a retrospective cohort study, 94 infants with stage 4S NB received a variety of treatments, e.g., CO regimen (CY and vincristine [VCR]), CE regimen (CBDCA and ETP), CADO regimen (CY, VCR, and DXR), ETP–cisplatin regimen, etc. [3]. In these studies, DXR was the anthracycline (AC) used [2,3]. However, to the best of our knowledge, there are no reports on the use of pirarubicin (THP) as an alternative AC in the treatment of a 1-month-old infant during the diagnosis of stage MS NB. THP is a synthetic analogue of DXR and was discovered and developed in Japan [7].

We modified the low-dose chemotherapy performed for infants with INSS stage III NB without *MYCN* amplification (intermediate-risk NB in the INRG classification system) used successfully in a previous study [8,9]. This modified chemotherapy regimen was used to treat a boy who was diagnosed with stage MS low-risk NB in the INRG classification system at the age of 42 days [9]. The chemotherapy used in his treatment (CY and THP) appeared to be effective and safe.

We present the following case in accordance with the CARE guidelines (http://www.care-statement.org, accessed on 11 February 2023).

## 2. Case Presentation

A 42-day-old boy was admitted to our hospital with hepatomegaly. His medical and family history were unremarkable.

On physical examination he was alert, not in a good general condition, but stable. His vital signs were as follows: heart rate, 135 beats/min; blood pressure, 120/90 mmHg; respiratory rate, 30 breaths/min; and oxygen saturation, 97%. His lungs were clear, his abdomen distended, and his liver palpable, 7 cm below the right costal margin.

The laboratory findings are shown in Table 1.

Laboratory analysis revealed elevated levels of blood catecholamines and tumor markers (neuron-specific enolase [NSE], urinary vanillylmandelic acid [u-VMA], and urinary homovanillic acid [u-HVA]). The imaging findings are shown in Figure 1. Radiography revealed massive hepatomegaly (Figure 1E). Ultrasonography showed a 33 mm × 22 mm right adrenal enlargement and multiple masses in the liver. Contrast-enhanced computed tomography (CT) showed a well-circumscribed nodule, 26 mm in diameter, in the right adrenal gland (Figure 1F). ^123^I-MIBG scintigraphy showed high uptake in the liver, but no abnormal uptake at other sites. BM examination revealed BM metastasis and the percentage of tumor cells was 3.6%. Based on these findings, the patient was staged as 4S according to the INSS and MS using the INRG Staging System [4,5]. On examination of his liver tumor biopsy sample, a pathological diagnosis of “poorly differentiated neuroblastoma with low mitosis-karyorrhexis index, favorable histology” was made based on the International Neuroblastoma Pathology Classification [10]; his tumor cell ploidy status was hyperdiploid, and MYCN was not amplified. According to these findings, his tumor was considered low-risk.

Figure 1 shows the clinical course.

He later showed signs of respiratory distress caused by rapidly evolving hepatomegaly. Contrast-enhanced CT showed increased hepatomegaly (Figure 1G); therefore, treatment was indicated. After obtaining written informed consent for chemotherapy from his family, we performed two cycles of Regimen A, the low-dose chemotherapy regimen administered in the previous study [8]. From the second week following admission, VCR 0.5 mg/m^2^ was administered on day 1 and CY 100 mg/m^2^ on day 8, but CT in the third week showed a further increase in hepatomegaly (Figure 1H). From the fourth week following admission, VCR 1.0 mg/m^2^ was administered on day 1 and CY 200 mg/m^2^ on day 8 [8]. Efficacy and adverse events were assessed after each cycle. In the sixth week following admission, his abdominal circumference had increased from 42.5 cm (at admission) to 49 cm and radiography showed massive hepatomegaly (Figure 1I); his tumors were evaluated as progressive disease according to the International Neuroblastoma Response Criteria [11]. We modified Regimen C2 (CY and VCR on day 1 and THP on day 3) used in the previous study and administered THP 15 mg/m^2^ on day 1 and CY 300 mg/m^2^ on day 8 (we assessed VCR as ineffective and excluded it from the regimen) [8]. After administration of THP and CY, the abdominal mass began to shrink and continued to do so; his tumor markers decreased substantially. His abdominal circumference decreased to 46 cm in the ninth week following admission, and then, in the tenth week, his tumor markers had decreased as follows: NSE 15.1 ng/mL, u-VMA 41 μg/mg Cre, and u-HVA 37 μg/mg Cre, and his adrenal tumor disappeared. Contrast-enhanced CT in week 13 showed reduced hepatomegaly (Figure 1J). In week 14 following admission, his abdominal circumference further decreased to 42 cm and his tumor markers were as follows: NSE 18.2 ng/mL, u-VMA 11.4 μg/mg Cre, and u-HVA 22.1 μg/mg Cre, and he was discharged. After discharge, there was no re-elevation of tumor markers or re-appearance of abdominal distension, and the hepatomegaly and liver metastases disappeared on contrast-enhanced CT after 1 year. During the 5-year follow-up, there was no re-elevation of tumor markers, his growth and development were normal, and he progressed without sequelae such as cardiac complications.

## 3. Discussion

This case report presents two important clinical suggestions. First, the regimen including THP appears to have been effective for an infant diagnosed with stage MS low-risk NB at the age of 1 month. Asymptomatic infants with stage 4S NB can be followed up with supportive care only, but they require cytotoxic therapy if they are symptomatic or are at risk of complications [2,3]. Our patient was considered to have an impending organ impairment; he showed respiratory distress caused by rapidly evolving hepatomegaly, and treatment was therefore indicated. Although we did not examine numeric chromosome aberrations and segmental chromosome aberrations of his tumor, which could have been associated with a worse outcome in NB [12,13], we examined him based on other features of the INRG classification system [9]. These included his pathological diagnosis on the International Neuroblastoma Pathology Classification: poorly differentiated neuroblastoma with low mitosis–karyorrhexis index, favorable histology; tumor cell ploidy status: hyperdiploid; and *MYCN* amplification: not amplified. Based on these factors, we determined that his tumor was low-risk and that low-dose chemotherapy was necessary. Thus, using the low-dose regimen performed for infants with INSS stage III neuroblastoma without *MYCN* amplification (intermediate-risk NB in the INRG classification system) in the previous study [8,9], we administered two cycles of Regimen A. However, because the chemotherapy did not shrink his tumors, we administered a modified Regimen C2 of THP and CY (we assessed VCR as ineffective and excluded it from the regimen), and his condition began to improve. In concurrence with the previous study [8], we reduced the dosages according to his age at treatment initiation. We selected THP as the AC considering its superior antitumor efficacy and lower cardiotoxicity [14,15,16,17,18,19,20,21,22]. In the retrospective cohort study mentioned above [3], 57 infants who had stage 4S NB and had life-threatening symptoms received radiation therapy or chemotherapy, but their 3-year overall survival was only 80%; 20% of the infants still died. There were 11 deaths in the 41 infants who received the CO regimen (CY, 5 mg/kg/day, days 1–3; VCR, 0.05 mg/kg/day, day 1) as first-line therapy, whereas there were 0 deaths in the 6 infants who received the CE regimen (CBDCA, 6.6 mg/kg/day, days 1–3; ETP, 5 mg/kg/day, days 1–3) and 1 death (who received the CO regimen as second-line therapy) in the 10 infants who received hepatic irradiation (a total dose of 4.5 Gy administered in three doses of 1.5 Gy on three consecutive days). The 6 infants who received the CE regimen required only two chemotherapy cycles. This suggests that the 20% of the infants who died and those who receive more intensive treatments could have had better outcomes and avoided further intensive treatment if a more effective therapy was chosen [3]. In addition, in the prospective study mentioned above [2], 49 infants with stage 4S NB have received chemotherapy, which was administered for Intermediate-Risk NB in another prospective study; this includes CBDCA, ETP, CY, and DXR [2,6]. However, their 3-year survival rate was 81%, and 9 infants died [2]. If we classify our patient using the treatment group assignment of this study [2], he would be assigned to group 2 and receive the two cycles of chemotherapy (Cycle 1: CBDCA, 560 mg/m^2^, day 1; ETP, 120 mg/m^2^/day, days 1–3 and Cycle 2: CBDCA, 560 mg/m^2^, day 1; CY, 1000 mg/m^2^, day 1; DXR, 30 mg/m^2^, day 1) [2,6]. Considering the likelihood of the development of second malignancy and the acute and long-term effects of toxicity on hearing, heart, and kidneys, the toxicity of this chemotherapy regimen may be considered higher than that of our chemotherapy regimen. Our low-intensity and minimum requirement chemotherapy regimen could buy time until spontaneous regression occurs. Thus, the regimen that includes THP could merit further study in the treatment of early infants with stage MS low-risk NB because it appears to have improved the outcome of our infant who was 1 month old at diagnosis. However, it is also possible that his tumor spontaneously regressed or that the chemotherapy with two cycles of VCR and CY had an effect. Thus, to examine the effects of regimens including THP, validation in a large number of cases is necessary.

Second, the regimen that included THP appeared to be safe for the infant who was 1 month old at diagnosis. In an animal study in which eight ACs including THP and DXR were administered to golden hamsters whose cardio- and skin toxicity were evaluated, THP was the least toxic—in particular, it was the least cardiotoxic AC [14]. Furthermore, regarding acute toxicity, in a retrospective study that compared 459 patients with non-Hodgkin’s lymphoma (NHL) receiving THP-COP: THP, CY, VCR, and prednisolone or CHOP (CY, DXR, VCR, and prednisolone), the THP-COP group had significantly fewer cases of alopecia and gastrointestinal toxicities; arrhythmia tended to decrease, especially in older adult patients [15]. In a randomized comparative trial of modified CHOP (two thirds dosage) versus THP-COP versus THP-COPE (THP-COP + ETP) in 486 NHL patients, a decrease in ejection fraction of less than 40% was reported in seven patients, all of whom were in the CHOP group [16]. In addition, regarding late cardiotoxicity in 61 patients with acute lymphoblastic leukemia (ALL) [age: 7.6–25.7 years; follow-up time (years): median 8.1, range 1.7–12.5; ACs used: DXR and THP; total dose of ACs converted to THP (mg/m^2^): median 207, range 135–812], no significant cardiac dysfunction was detected in patients who received THP [17]. Furthermore, in a prospective study in 276 patients with childhood ALL, the THP arm was superior to the daunorubicin arm for late cardiotoxicity [18]. The selection of THP as the AC may therefore benefit patients in terms of acute and late toxicity, especially cardiotoxicity. However, validation in a large number of cases is also necessary to examine the safety of the regimens that include THP.

THP has several advantages over DXR. In terms of uptake into tumor cells, in an experiment using cultured mouse lymphoma L5178Y cells, the uptake velocity of THP was 170 times faster than that of DXR. In the inhibition of nucleic acid synthesis, the half-maximal inhibitory concentrations of THP and DXR on DNA synthesis were 0.1 μg/mL and 4.2 μg/mL, respectively, whereas those of THP and DXR on RNA synthesis were 0.23 μg/mL and 6.6 μg/mL, respectively. Therefore, the concentrations of THP were much lower than those of DXR [19]. In drug–tissue transfer, in an experiment in which mice received intravenous injections of the drugs, the plasma disappearance and tissue transfer of THP were faster than those of DXR and drug levels decreased more quickly in most tissues with THP use than with DXR [20]. With regard to the efficacy against experimental mouse tumors, THP was more effective against L1210 leukemia, Lewis lung carcinoma, B16 melanoma, and colon adenocarcinoma 38, and was more antimetastatic against Lewis lung carcinoma than DXR [21]. In DXR-resistant tumor cells, in the chemosensitivities of 12 P-glycoprotein-positive breast cancer specimens, THP was significantly more effective than DXR and epirubicin (EPI) [22], suggesting that THP may have different antitumor spectra from DXR and EPI. In summary, compared with DXR and EPI, THP has advantages in uptake into tumor cells [19], inhibition of nucleic acid synthesis [19], tissue transfer [20], antitumor efficacies [21], antimetastatic effects [21], and antitumor spectrum [22], suggesting its superior clinical utility. Although chemotherapies including DXR have been administered for unresectable low-risk NB in previous studies [2,23], chemotherapy that includes THP may be more beneficial to patients because of its superior antitumor efficacy and lower cardiotoxicity. In addition, an improved drug, hydroxypropyl-acrylamide polymer-conjugated THP (P-THP), is being developed [24]. Thus, simply replacing other ACs with THP or P-THP in a regimen may improve outcomes in AC-sensitive pediatric cancers. However, as in the above-mentioned comparison in childhood ALL [18], a direct comparison of the efficacy and safety of THP or P-THP with that of other ACs has not yet been examined in NB or other pediatric cancers. Further investigation is therefore necessary.

## 4. Conclusions

This case report presents two important clinical suggestions. The regimen including THP appears to have been effective and safe for an infant diagnosed with stage MS low-risk NB at 1 month of age. As THP may have superior antitumor efficacy and lower acute and late toxicity, especially cardiotoxicity, than DXR, a regimen that includes THP could merit further study in the treatment of stage MS low-risk NB during early infancy.

## Figures and Tables

**Figure 1 children-10-00871-f001:**
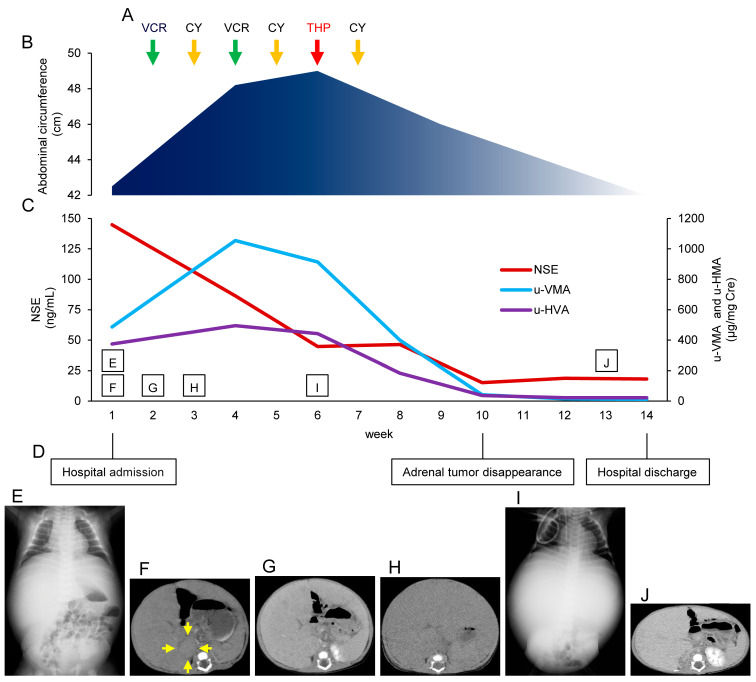
Changes during hospitalization: (**A**) Chemotherapy: letters and arrows indicate anticancer drugs and administration timings. Green arrows: VCR, at weeks 2 and 4; orange arrows: CY, at weeks 3, 5, and 7; red arrows: THP, at week 6. (**B**) Changes in abdominal circumference: this area graph represents the changes in abdominal circumference during hospitalization. The abdominal circumference increased from 42.5 cm at admission to 49 cm at week 6, and decreased to 46 and 42 cm at weeks 9 and 14, respectively. (**C**) Changes in tumor markers: this line graph represents the changes in tumor markers during hospitalization. Red line: NSE; violet line: u-VMA; blue line: u-HVA. (**D**) Important time points: week of hospital admission: week 1; week when the adrenal tumor disappeared: week 10; week of hospital discharge: week 14. (**E**) Chest-abdominal radiograph on admission: AP radiograph. Radiography revealed massive hepatomegaly. (**F**) An abdominal contrast-enhanced CT image on admission: a well-circumscribed nodule 26 mm in diameter was detected in the right adrenal gland. Yellow arrows indicate the nodule. (**G**) An abdominal contrast-enhanced CT image on the second week: the CT showed increased hepatomegaly. (**H**) An abdominal CT image in the third week: the CT showed additionally increased hepatomegaly. (**I**) Chest-abdominal radiograph in the sixth week: radiography revealed massive hepatomegaly. (**J**) An abdominal contrast-enhanced CT image in week 13: the CT showed reduced hepatomegaly. VCR: vincristine; CY: cyclophosphamide; THP: pirarubicin; NSE: neuron-specific enolase; u-VMA: urinary vanillylmandelic acid; u-HVA: urinary homovanillic acid; AP: anterior-posterior; CT: computed tomography.

**Table 1 children-10-00871-t001:** Laboratory data upon admission.

Variable	Value (Normal Range)
White blood cell count (cells/μL)	7100
Hemoglobin (g/dL)	8.3
Platelet count (platelets/μL)	226,000
International normalized ratio of prothrombin time	1.43
Activated partial-thromboplastin time (s)	36.5
Fibrinogen (mg/dL)	73
Fibrin degradation products (μg/mL)	5.4
Antithrombin-III (%)	63
Alanine aminotransferase (U/L)	11
C-reactive protein (mg/dL)	0.04
Dopamine (pg/mL)	68 (<30)
Adrenaline (pg/mL)	134 (100–400)
Noradrenaline (pg/mL)	1228 (<70)
Neuron-specific enolase (ng/mL)	144.8 (<16.3)
Urinary vanillylmandelic acid (μg/mg Cre)	487.1 (6–11)
Urinary homovanillic acid (μg/mg Cre)	374.2 (11–20)

Normal range: facility reference value.

## Data Availability

Not applicable.

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
