# Peer review of "Pirarubicin Combination Low-Dose Chemotherapy for Early Infantile Stage MS Neuroblastoma: Case Report"

_children, 2023, doi:10.3390/children10050871_

Round 1

Reviewer 1 Report

Dear Authors

Dear Authors

Your article is very interesting.

I have some questions:

1. Introduction - I don't understand - ....is a tumor which is diagnosed in infancy in 40% of cases. Could you explain it?

line 52 - what doses of the liver radiation were used?

line 55-57 - in which cases high dose chemotherapy was used?

2. Case presentation

Table 1 - it is necessary to include references

did you assess NCA and SCA in the tumor cells?

why weren't you used CARBOPLATIN AND ETPOPOSIDE in this case?

3. Discussion

What is prognosis in this cases?

Are there the current clinical studies concerning treatment of infants with neuroblastoma?

What is a current standard of treatment for infants with neuroblastomas?

Author Response

Responses to Reviewer 1 Comments:

              We are grateful to Reviewer 1 for the critical comments and useful suggestions that have helped us to improve our paper. As indicated in the responses that follow, we have taken all of these comments and suggestions into account in the revised version of the paper.

#. Your article is very interesting.

Response: Thank you very much for your comment. The patient was not in good general condition but was stable. He was later considered to be at risk of impending organ impairment when he showed signs of respiratory distress caused by rapidly evolving hepatomegaly, and treatment was therefore initiated. We modified and administered existing regimens used successfully in the treatment of intermediate-risk neuroblastoma (NB) in Japan. The regimens were of low-intensity and minimum requirement because spontaneous regressions are sometimes observed in Stage MS NB.

Messages: Pirarubicin combination low-dose chemotherapy for early infantile stage MS low-risk NB could merit further study because of its low toxicity.

The administration of up-front, low-dose chemotherapy could possibly be merited in the treatment of stage MS low-risk NB when there are concerns of exacerbation or complication.

#1. Introduction

#1-1. I don't understand - ....is a tumor which is diagnosed in infancy in 40% of cases. Could you explain it?

Response: Thank you for your valuable question. We mean that approximately 40% of NB patients are diagnosed in infancy.

#1-2. line 52 - what doses of the liver radiation were used?

Response: Thank you for your valuable question. We have withdrawn this reference [Nickerson, H.J.; et al. Favorable biology and outcome of stage IV-S neuroblastoma with supportive care or minimal therapy: a Children’s Cancer Group study. J Clin Oncol. 2000, 18, 477–486.], because there is a more recent report written by the Children’s Oncology Group (COG) [Twist, C.J.; et al. Defining risk factors for chemotherapeutic intervention in infants with stage 4S neuroblastoma: A report from Children's Oncology Group Study ANBL0531. J Clin Oncol. 2019, 37, 115–224. DOI: 10.1200/JCO.18.00419.]. We have cited the more recent report in the revised manuscript instead. However, the dosage of hepatic radiation used by Nickerson et al. was 4.5 Gy over 3 days.

#1-3. line 55-57 - in which cases high dose chemotherapy was used?

Response: Thank you for your valuable question. We have deleted the reference to high dose chemotherapy because it is not routinely used in the treatment of stage MS NB and is therefore not pertinent to this paper.

#2. Case presentation

#2-1. Table 1 - it is necessary to include references

Response: Thank you for your valuable comment. We regret the confusion. We actually had a discussion with the English editing company, Editage, regarding this prior to submission. Since the definition of pediatric is 0–15 years, there is no consistent normal range of laboratory values. We have removed the items of low importance from the table and retained only the most relevant items for consideration in the revised manuscript, so that the table does not become unnecessarily busy. Our facility reference values for tumor markers have now been added (Table 1, lines 79–80, yellow).

#2-2. Did you assess Numerical chromosomal aberrations (NCA) and segmental chromosomal aberration (SCA) in the tumor cells?

Response: Thank you for your valuable question. We did not assess these chromosomal aberrations (lines 153–154, blue).

#2-3. Why weren't you used CARBOPLATIN AND ETPOPOSIDE in this case?

Response: Thank you for this important question. In Japan, the treatment of stage MS NB is not standardized, and the treatment policy is decided according to the individual case. We modified the low-dose chemotherapy regimen administered to infants with INSS stage â…¢ NB without MYCN amplification [intermediate-risk NB on the International Neuroblastoma Risk Group (INRG) classification system] in a previous prospective study to treat our patient who was 42 days old at diagnosis of stage MS low-risk NB according to the INRG classification system (lines 63–66, blue and yellow). That is, we used “tried and tested” anticancer drugs and regimens that had been used successfully in similar situations. [Iehara, T.; et al.; Japanese Infantile Neuroblastoma Cooperative Study Group. Successful treatment of infants with localized neuroblastoma based on their MYCN status. Int J Clin Oncol. 2013, 18, 389–395. DOI:10.1007/s10147-012-0391-y.] [Cohn, S.L.; et al. The International Neuroblastoma Risk Group (INRG) classification system: an INRG Task Force report. J Clin Oncol. 2009, 27, 289–297. DOI:10.1200/JCO.2008.16.6785.]

#3. Discussion

#3-1. What is prognosis in this cases?

#3-2. Are there the current clinical studies concerning treatment of infants with neuroblastoma?

#3-3. What is a current standard of treatment for infants with neuroblastomas?

Response: Thank you for these valuable questions. We have now cited the more recent COG report [Twist, C.J.; et al. Defining risk factors for chemotherapeutic intervention in infants with stage 4S neuroblastoma: A report from Children's Oncology Group study ANBL0531. J Clin Oncol. 2019, 37, 115–224. DOI: 10.1200/JCO.18.00419.].

(#3-1) In this report, the 3-year overall survival of infants with stage 4S NB was 81.4% ± 5.8%; our patient progressed without sequelae such as cardiac complications and his growth and development were normal (lines 144–145, blue). More definite prognoses will only be determined after further follow-up.

(#3-2) Yes, to our knowledge, for stage 4S NB, studies are only being conducted by the COG.

(#3-3) Treatment is currently not standardized in Japan. In the study we referred to, the regimens contained cyclophosphamide, vincristine, cisplatin, and pirarubicin in various combinations [Iehara, T.; et al.; Japanese Infantile Neuroblastoma Cooperative Study Group. Successful treatment of infants with localized neuroblastoma based on their MYCN status. Int J Clin Oncol. 2013, 18, 389–395. DOI:10.1007/s10147-012-0391-y.]. In one previous study, using the International Neuroblastoma Pathology Classification, MYCN copy number, DNA index, and loss of heterozygosity at 1p36 and 11q23, patients with stage 4S NB were divided into 4 groups, and chemotherapy previously used to treat intermediate-risk NB was administered to each group. The regimens contained Carboplatin, Etoposide, Cyclophosphamide, and Doxorubicin [Twist, C.J.; et al. Defining risk factors for chemotherapeutic intervention in infants with stage 4S neuroblastoma: A report from Children's Oncology Group study ANBL0531. J Clin Oncol. 2019, 37, 115–224. DOI: 10.1200/JCO.18.00419.] [Baker, D.L.; et al. Outcome after reduced chemotherapy for intermediate-risk neuroblastoma. N Engl J Med. 2010, 363, 1313–1323. DOI: 10.1056/NEJMoa1001527].

Reviewer 2 Report

MS neuroblastoma with Life Threatening Symptoms should be treated with up-front chemotherapy.

Low risk Ms neuroblastoma occurs in infant and presents a very good long terme prognosis.

this report shows a case of MS NB without initial response to vincristine and low doses cyclophosphamide. The treatment was switched to Vincristine Cyclophosphamide and THP (Pirarubicine) with further regression and cure of the child.

Even, this association was efficient, we can deplore that:

- there is no phase I and II of pirarubicine published in neuroblastoma and this drug seems to be used only in Japan.

-even Pirarubicine seems efficient, it remains an anthracycline with long term cardiotoxicity particularly in a an infant.

Association of Carboplatinum and Vepeside seems efficient and with less toxicity in this situation.

The paper is too long considering the imput of these datas.

Author Response

Responses to Reviewer 2 Comments:

              We are grateful to Reviewer 2 for the critical comments and useful suggestions that have helped us to improve our paper. As indicated in the responses that follow, we have taken all of these comments and suggestions into account in the revised version of the paper.

#. MS neuroblastoma (NB) with Life Threatening Symptoms should be treated with up-front chemotherapy. Low risk MS NB occurs in infant and presents a very good long terme prognosis. This report shows a case of MS NB without initial response to vincristine and low doses cyclophosphamide. The treatment was switched to Vincristine Cyclophosphamide and THP (Pirarubicine) with further regression and cure of the child.

Response: Thank you very much for your comment. The patient was not in good general condition but was stable. He was later considered to be at risk of impending organ impairment when he showed signs of respiratory distress caused by rapidly evolving hepatomegaly, and treatment was therefore initiated. We modified and administered existing regimens used successfully in the treatment of intermediate-risk neuroblastoma (NB) in Japan. The regimens were of low-intensity and minimum requirement because spontaneous regressions are sometimes observed in Stage MS NB.

Messages: THP combination low-dose chemotherapy for early infantile stage MS low-risk NB could merit further study because of its low toxicity.

The administration of up-front, low-dose chemotherapy could possibly be merited in the treatment of stage MS low-risk NB when there are concerns of exacerbation or complication.

#1. There is no phase I and II of pirarubicine (THP) published in NB and this drug seems to be used only in Japan.

Response: Thank you for your valuable statement. According to information from two companies selling this drug, THP is also used in the treatment of NB (e.g., DOI: 10.1155/2022/8319221) and several other cancers, in China. In addition, an improved drug, hydroxypropyl-acrylamide polymer-conjugated THP, is being developed [Makimoto, A; et al. Development of a selective tumor-targeted drug delivery system: hydroxypropyl-acrylamide polymer-conjugated pirarubicin (P-THP) for pediatric solid tumors. Cancers (Basel). 2021, 13. DOI:10.3390/cancers13153698.] (lines 238–239, blue). Thus, in future, the use of hydroxypropyl-acrylamide polymer-conjugated THP and THP in the treatment of NB and other cancers may be expanded to several other countries. We have added a description of THP to the Introduction section of the revised manuscript for completeness [Umezawa, H.; et al. Tetrahydropyranyl derivatives of daunomycin and adriamycin. J Antibiot (Tokyo). 1979, 32, 1082–1084. DOI: 10.7164/antibiotics.32.1082.] (lines 61–62, gray).

#2. Even THP seems efficient, it remains an anthracycline (AC) with long term cardiotoxicity particularly in an infant.

Response: Thank you for your valuable comment. As you have pointed out, THP may have long-term cardiotoxicity; however, to our knowledge it is the least cardiotoxic AC. We regret the confusion. Although THP is an anthracycline, the selection of THP as the AC may benefit patients especially in terms of cardiotoxicity (lines 214–215, blue). A statement regarding cardiotoxicity is included in the Discussion (paragraph 2). In addition, we have included a reference to a prospective study in childhood acute lymphoblastic leukemia. In brief, for late cardiotoxicity (BNP, FS, QTc), THP was found to be superior to daunorubicin (DNR) [Hori, H.; et al. Acute and late toxicities of pirarubicin in the treatment of childhood acute lymphoblastic leukemia: results from a clinical trial by the Japan Association of Childhood Leukemia Study. Int J Clin Oncol. 2017, 22, 387–396. DOI:10.1007/s10147-016-1062-1.2(2):387-396.] (lines 212–214, gray). We would appreciate it if you could take a look. In Japan, considering the long-term toxicities (such as auditory disorder, nephrotoxicity, second malignancy) associated with carboplatin (CBDCA) and Vepesid: etoposide (ETP), physicians have refrained from using them in the treatment of stage MS low-risk NB.

#3. Association of CBDCA and ETP seems efficient and with less toxicity in this situation.

Response: Thank you for your valuable comment. We agree that CBDCA and ETP may be more efficient in this situation. However, in Japan, the use of CBDCA and ETP in the treatment of stage MS low-risk NB has been avoided, considering their long-term toxicities (auditory disorder, nephrotoxicity, second malignancy, etc.). The treatment of stage MS NB is not standardized in Japan, and the treatment policy is decided according to the individual case. Thus, we modified the low-dose chemotherapy administered to infants with INSS stage â…¢ NB without MYCN amplification (intermediate-risk NB in the INRG classification system) that was used successfully in a previous prospective study [Iehara, T.; et al.; Japanese Infantile Neuroblastoma Cooperative Study Group. Successful treatment of infants with localized neuroblastoma based on their MYCN status. Int J Clin Oncol. 2013, 18, 389–395. DOI:10.1007/s10147-012-0391-y.] [Cohn, S.L.; et al. The International Neuroblastoma Risk Group (INRG) classification system: an INRG Task Force report. J Clin Oncol. 2009, 27, 289–297. DOI:10.1200/JCO.2008.16.6785.], and using this modified chemotherapy regimen, treated our patient who was 42 days old at diagnosis of stage MS low-risk NB according to the INRG classification system (lines 63–66, blue and yellow).

#4. The paper is too long considering the input of these data.

Response: Thank you for your valuable comment. We agree that this paper was too long. We have removed redundant sections and have edited some sentences for improved clarity and flow.

Reviewer 3 Report

The authors present a single case report of novel use of the chemotherapy pirarubicin, a modified variant of doxorubicin, in an infant who presented with stage MS neuroblastoma with progressive abdominal girth size. The initial premise - that pirarubicin was safely and effectively added to treatment for a single patient with clinically progressive neuroblastoma - is somewhat interesting. However, the paper has the following major challenges:

1) Pirarubicin does not seem to be widely available outside of Japan. It is not commercially available in the US for any treatments. This dramatically reduces the potential impact of the work. 

2) At the doses used (a single dose of pirarubicin), an equivalent dose of doxorubicin would be expected to have clinically efficacy without significant toxicity. 

3) Stage MS neuroblastomas with favorable histology are known to spontaneously regress. While this patient had significant abdominal distension at time of presentation, it is not clear that the child was actually "symptomatic" of his or her disease. Accepting that chemotherapy is reasonable for such a case, the authors never actually make a case to show that the tumor did not simply start to regress, either after initial response to pirarubicin or independently of treatment. 

4) The authors simply cannot say a regimen using pirarubicin can be an option in the treatment of early infants with stage MS low-risk disease. At best, they can suggest that this combination merits further study in this population or in other pediatric cancers. Treatment based on a single case report is not sufficient to justify broader use without validation in some type of larger clinical trial. 

Below are additional critiques that will improve the flow and accuracy of the manuscript. However, with the issues above, the manuscript is not ready for publication. Significant modifications may improve the manuscript to allow publication, but its utility will remain low because of the lack of availability of pirarubicin outside of Japan. 

The first sentence is not particularly informative. Just stating that neuroblastoma is a malignant tumor leaves much to be desired. There should be a brief reminder of the tissue of origin, that it is the most common tumor of infancy, but that it is generally associated with good outcomes. A concise statement of the unusual nature of Stage MS (including the section in paragraph 2) can then be made. There should also be clarification of what "low-risk" and "good prognosis" mean, as that can vary from disease to disease. 

The authors should make it clearer that stage MS and 4S are the same disease, just using different classification systems. 

The authors state that patients with stage MS are usually treated with "chemo-, radio-, and surgical therapy." Many patients are actually observed when there are not acute symptoms, and studies are ongoing to better delineate the natural course of this disease process. It would be very unusual to use radiation therapy in these patients, given the severe risk of toxicity in infants and generally good benefit with chemotherapy and surgery alone if needed. 

The citations the authors use referencing radiation therapy and high-dose chemotherapy are twenty years old or older and not particularly relevant. 

The introduction also implies that there is no recommended initial approach for chemo for patients with symptomatic stage MS disease; it is unclear what the standard approach may be in Japan, but dosing of chemotherapy in infants has been more standardized in the US, using ANBL0531 as one backbone.There are publications on this protocol in general and for Stage 4S neuroblastoma specifically. That regimen used carboplatin and etoposide in the first cycle, and then carboplatin, cyclophosphamide and doxorubicin. 

In the introduction, the authors say that the regimen of vincristine, cyclophosphamide and pirarubicin is "effective and safe," but they did not actually use vincristiine with cyclophosphamide/pirarubicin. That should be clarified. 

In the Case presentation, there is an almost excessive amount of data provided. The full details of initial vital signs are not really necessary or informative, as tachypnea and hypertension are the only pertinent features. The full list of laboratory values are also not necessary, particularly without a normal range. A narrowed list of pertinent normal and abnormal values would suffice. Similarly, having the X-ray, ultrasound, CT and MIBG scan are redundant, as are the photomicrographs of the bone marrow and tumor. As the main idea of the manuscript is response to therapy, there does not need to be an endless detailing of the patient presentation. 

On line 126, the authors reference treatment of Regimen A "from the previous study." As many studies were previously cited, it would be best to include a citation again here. 

The treatment course is somewhat cryptic. Originally chemotherapy was being administered as single agents, one week at a time, then suddenly stopped when the abdominal girth start to shrink. Why was it stopped at that point, vs ongoing treatment? This is where showing the repeat imaging would be far more informative. 

In follow-up, the authors report that the abd circumference slowly decreased, lagging behind the urinary markers, but then report the lack of liver metastases at 1 year post treatment. Did they really not assess the tumor radiographically until then?

In the discussion, the benefits of THP are discussed in other cancers. While there is potential benefit over doxorubicin in this way, both in efficacy and safety, those results are never demonstrated or discussed in neuroblastomas, either preclinically or clinically. 

Publications have shown that the combination of vincristine, cyclophosphamide and doxorubicin are specifically efficacious against neuroblastomas, with a specific mechanistic sequence. As such, it is not clear why the authors chose to exclude vincristine from ongoing treatment. 

The authors reference the CARE reporting checklist. That should be explained and a citation provided, as many readers will not know what that is. 

Author Response

Responses to Reviewer 3 Comments:

              We are grateful to Reviewer 3 for the critical comments and useful suggestions that have helped us to improve our paper. As indicated in the responses that follow, we have taken all of these comments and suggestions into account in the revised version of the paper.

#. The authors present a single case report of novel use of the chemotherapy pirarubicin (THP), a modified variant of doxorubicin (DXR), in an infant who presented with stage MS neuroblastoma (NB) with progressive abdominal girth size. The initial premise - that pirarubicin was safely and effectively added to treatment for a single patient with clinically progressive NB - is somewhat interesting.

Response: Thank you very much for your comment. The patient was not in good general condition but was stable. He was later considered to be at risk of impending organ impairment when he showed signs of respiratory distress caused by rapidly evolving hepatomegaly, and treatment was therefore initiated. We modified and administered existing regimens used successfully in the treatment of intermediate-risk neuroblastoma (NB) in Japan. The regimens were of low-intensity and minimum requirement because spontaneous regressions are sometimes observed in Stage MS NB.

Messages: THP combination low-dose chemotherapy for early infantile stage MS low-risk NB could merit further study because of its low toxicity.

The administration of up-front, low-dose chemotherapy could possibly be merited in the treatment of stage MS low-risk NB when there are concerns of exacerbation or complication.

#1. THP does not seem to be widely available outside of Japan. It is not commercially available in the US for any treatments. This dramatically reduces the potential impact of the work.

Response: Thank you for your valuable comment. According to information from two companies selling this drug, THP is also used in the treatment of NB (e.g., DOI: 10.1155/2022/8319221) and several other cancers in China. In addition, an improved drug, hydroxypropyl-acrylamide polymer-conjugated THP (P-THP), is being developed [Makimoto, A.; et al, H. Development of a selective tumor-targeted drug delivery system: P-THP for pediatric solid tumors. Cancers (Basel). 2021, 13. DOI:10.3390/cancers13153698.] (lines 238–239, blue). Thus, in future, the use of P-THP and THP in the treatment of NB and other cancers may be expanded to several other countries. We have added a description of THP to the Introduction section of the revised manuscript for completeness [Umezawa, H.;et al. Tetrahydropyranyl derivatives of daunomycin and adriamycin. J Antibiot (Tokyo). 1979, 32, 1082–1084. DOI: 10.7164/antibiotics.32.1082.] (lines 61–62, gray).

#2. At the doses used (a single dose of THP), an equivalent dose of DXR would be expected to have clinically efficacy without significant toxicity.

Response: Thank you for your valuable comment. I am very happy to receive a comment that suggests that you read the second paragraph of the Discussion with interest. I hope that this content appropriate.

#3. Stage MS NB with favorable histology are known to spontaneously regress. While this patient had significant abdominal distension at time of presentation, it is not clear that the child was actually "symptomatic" of his or her disease. Accepting that chemotherapy is reasonable for such a case, the authors never actually make a case to show that the tumor did not simply start to regress, either after initial response to THP or independently of treatment.

Response: Thank you for bringing this to our attention. We have inserted “He later showed signs of respiratory distress caused by rapidly evolving hepatomegaly. Contrast-enhanced CT showed increased hepatomegaly (Figure 1G); therefore treatment was indicated.” (lines 118–120, green) and “; he showed respiratory distress” (line 152, green). We regret the confusion. We agree that his tumor might have spontaneously regressed or that the chemotherapy with two cycles of vincristine (VCR) and cyclophosphamide (CY) may have had an effect (lines 193–195, blue). In light of these possibilities, we have amended the first paragraph of the Discussion. We would appreciate it if you could take a look.

#4. The authors simply cannot say a regimen using THP can be an option in the treatment of early infants with stage MS low-risk disease. At best, they can suggest that this combination merits further study in this population or in other pediatric cancers. Treatment based on a single case report is not sufficient to justify broader use without validation in some type of larger clinical trial.

Response: Thank you for your valuable comment. We revised the text to “A regimen that includes pirarubicin could merit further study (be used) in the treatment of early infants with stage MS low-risk NB who are at risk of complications (line 34, green)”, “the regimen that includes THP could merit further study (be used as an effective option) in the treatment of early infants with stage MS low-risk NB (line 191, green)”, and “a regimen that includes THP could merit further study (be used as one of the options) in the treatment of stage MS low-risk NB during early infancy (line 249, green)

#5. The first sentence is not particularly informative. Just stating that NB is a malignant tumor leaves much to be desired. There should be a brief reminder of the tissue of origin, that it is the most common tumor of infancy, but that it is generally associated with good outcomes. A concise statement of the unusual nature of Stage MS (including the section in paragraph 2) can then be made. There should also be clarification of what "low-risk" and "good prognosis" mean, as that can vary from disease to disease.

Response: Thank you for your valuable comment. We have inserted “neural crest derived” (lines 21 and 39, green), “and it is generally associated with good outcomes in infants,” (line 41, green), and “in the INRG classification system [Cohn, S.L.; et al. The International Neuroblastoma Risk Group (INRG) classification system: an INRG Task Force report. J Clin Oncol. 2009, 27, 289–297. DOI:10.1200/JCO.2008.16.6785.]” (lines 66–67, green). We have deleted “low-risk” in “Stage MS NB is usually low-risk with has a good prognosis.” (line 51, green).

#6. The authors should make it clearer that stage MS and 4S are the same disease, just using different classification systems.

Response: Thank you for your valuable comment. We have edited the text to “Stage MS NB classified using the International Neuroblastoma Risk Group (INRG) Staging System is a metastatic disease, usually diagnosed in patients younger than 18 months [Monclair, T.; et al. The International Neuroblastoma Risk Group (INRG) staging system: an INRG Task Force report. J Clin Oncol. 2009, 27, 298–303. DOI:10.1200/JCO.2008.16.6876.]. The same disease is classified as stage 4S NB (younger than 12 months) using the International Neuroblastoma Staging System (INSS) [Brodeur, G.M.; et al. International criteria for diagnosis, staging, and response to treatment in patients with neuroblastoma. J Clin Oncol. 1988, 6, 1874–1881. DOI:10.1200/JCO.1988.6.12.1874].” (lines 44–47, green).

#7. The authors state that patients with stage MS are usually treated with "chemo-, radio-, and surgical therapy." Many patients are actually observed when there are not acute symptoms, and studies are ongoing to better delineate the natural course of this disease process. It would be very unusual to use radiation therapy in these patients, given the severe risk of toxicity in infants and generally good benefit with chemotherapy and surgery alone if needed.

Response: Thank you for your valuable comment. We have inserted “followed up regularly” and deleted “radio-” (line 52, green).

#8. The citations the authors use referencing radiation therapy and high-dose chemotherapy are twenty years old or older and not particularly relevant.

Response: Thank you for your valuable comment. We have deleted “hepatic irradiation” and “high dose chemotherapy (Buslphan-melphalan, BCNC-VM26-CBDCA, or CBDCA-melphalan) followed by autologous BM or peripheral stem cell rescue”, and inserted “etc” as we agree that these therapies may no longer be relevant (line 58, green).

#9. The introduction also implies that there is no recommended initial approach for chemo for patients with symptomatic stage MS disease; it is unclear what the standard approach may be in Japan, but dosing of chemotherapy in infants has been more standardized in the US, using ANBL0531 as one backbone. There are publications on this protocol in general and for Stage 4S NB specifically. That regimen used carboplatin (CBDCA) and etoposide (ETP) in the first cycle, and then CBDCA, CY and DXR.

Response: Thank you for your valuable comment. We deleted “No standard therapies exist” and revised it to “In a prospective study, 49 infants with stage 4S NB received chemotherapy with CBDCA, ETP, cyclophosphamide (CY), and doxorubicin (DXR) [Twist, C.J.; et al. Defining risk factors for chemotherapeutic intervention in infants with stage 4S neuroblastoma: A report from Children's Oncology Group study ANBL0531. J Clin Oncol. 2019, 37, 115–224. DOI: 10.1200/JCO.2008.16.6785.].” (lines 53–55, green). In the US there is a standard treatment regimen for this disease but there is no standard therapy regimen for Stage MS NB in Japan. However, there is a chemotherapy regimen for intermediate-risk NB in Japan. We used anticancer drugs and regimens that were familiar and had been used successfully in the past. Because spontaneous regressions are observed in Stage MS NB and a good outcome can be expected with low-intensity and minimum requirement chemotherapy, THP combination low-dose chemotherapy may merit further study, as you advised.

#10. In the introduction, the authors say that the regimen of VCR, CY and THP is "effective and safe," but they did not actually use VCR with CY/THP. That should be clarified.

Response: Thank you for mentioning this. We have deleted “VCR” for clarity (line 68, green).

#11. In the Case presentation, there is an almost excessive amount of data provided. The full details of initial vital signs are not really necessary or informative, as tachypnea and hypertension are the only pertinent features. The full list of laboratory values are also not necessary, particularly without a normal range. A narrowed list of pertinent normal and abnormal values would suffice. Similarly, having the X-ray, ultrasound, CT and MIBG scan are redundant, as are the photomicrographs of the bone marrow and tumor. As the main idea of the manuscript is response to therapy, there does not need to be an endless detailing of the patient presentation.

Response: Thank you for your valuable comment. We have deleted “height, weight, temperature” (line 74, green), “Total bilirubin, Direct bilirubin, Aspartate aminotransferase, Alkaline phosphatase, Total protein, Albumin, Creatinine, Lactate dehydrogenase, Ferritin” (Table 1, green), and “Figure 1B: Ultrasonography; Figure 1D: 123I-MIBG scintigraphy; Figure 1E: BM examination; Figure 1F: tumor biopsy” (Figure 1 and the legend, green).

#12. On line 126, the authors reference treatment of Regimen A "from the previous study." As many studies were previously cited, it would be best to include a citation again here.

Response: Thank you for your valuable comment. We have inserted “[8]” (line 122, green).

#13. The treatment course is somewhat cryptic. Originally chemotherapy was being administered as single agents, one week at a time, then suddenly stopped when the abdominal girth start to shrink. Why was it stopped at that point, vs ongoing treatment? This is where showing the repeat imaging would be far more informative.

Response: Thank you for your valuable comment. We have inserted Xp and CT images into the revised Figure 1. We apologize for the confusion. CY was administered in week 7 when the abdominal girth began to shrink, and chemotherapy was stopped as a result of confirming that his abdominal girth had continued to shrink in week 8. Of course, if the tumor had regrown, we planned to restart chemotherapy. We have inserted “contrast-enhanced CT showed increased hepatomegaly (Figure 1G)” (line 119, green), “but CT in the third week showed a further increase in hepatomegaly (Figure 1H)” (lines 123–124, green), “and radiography showed massive hepatomegaly (Figure 1I)” (line 128, green), “continued to do so” (line 133, green), and “Contrast-enhanced CT in week 13 showed reduced hepatomegaly (Figure 1J)” (lines 137–138, green), and revised the Figure 1 Legend (green).

#14. In the discussion, the benefits of THP are discussed in other cancers. While there is potential benefit over doxorubicin in this way, both in efficacy and safety, those results are never demonstrated or discussed in neuroblastomas, either preclinically or clinically.

Response: Thank you for your valuable comment. We have added “However, as in the above-mentioned comparison in childhood ALL [16], a direct comparison of the efficacy and safety of THP or P-THP with that of other ACs has not yet been examined in NB or other pediatric cancers. Further investigation is therefore necessary.” (lines 241–243, green). THP and DXR have been compared in childhood acute lymphoblastic leukemia [Hori, H.; et al. Acute and late toxicities of pirarubicin in the treatment of childhood acute lymphoblastic leukemia: results from a clinical trial by the Japan Association of Childhood Leukemia Study. Int J Clin Oncol. 2017, 22, 387–396. DOI:10.1007/s10147-016-1062-1.2(2):387-396.].

#15. Publications have shown that the combination of vincristine, cyclophosphamide and doxorubicin are specifically efficacious against neuroblastomas, with a specific mechanistic sequence. As such, it is not clear why the authors chose to exclude vincristine from ongoing treatment.  

Response: Thank you for your valuable comment. Again, we apologize for the confusion. We said, “because the chemotherapy did not shrink his tumors, we administered a modified Regimen C2 of THP and CY (we assessed VCR as ineffective and excluded it from the regimen)” (lines 163–165, blue). We have inserted “(we assessed VCR as ineffective and excluded it from the regimen)” (line 132, green). We thought it may be better to describe it in a manner that is easy for readers to understand, as an in-depth explanation would be redundant.

#16. The authors reference the CARE reporting checklist. That should be explained and a citation provided, as many readers will not know what that is.  

Response: Thank you for your valuable comment. We have inserted “[https://www.care-statement.org/]” (line 70, green).

Round 2

Reviewer 1 Report

Dear Authors

It is necessary to include in the article:

1. information about NCA and SCA

2. information about doses of radiotherapy of the liver

Reviewer 2 Report

OK with this revised version

Reviewer 3 Report

I appreciate the author's clear efforts to address my questions and concerns. The manuscript is considerably improved, use of THP better justified, review of the results better and more even-handed, and discussion more judicious. While I still feel that the application of THP over DXR in this case isn't fully justified (absence of preclinical data on neuroblastoma, and low-to-absent cardiotoxicity from this dosing of DXR), the paper could lead to greater research. Manuscript is acceptable, though impact will remain low due to the limited accessibility of the drug.